# SENSITIVITY SAMPLING FOR CORESET-BASED DATA SELECTION

## ABSTRACT

Given the sustained growth in both training data and model parameters, the problem of finding the most useful training data has become of primary importance for training state-of-the-art and next generation models.

We work in the context of data selection and consider the problem of finding the best representative subset of a dataset to train a machine learning model. Assuming embedding representation of the data (coming for example from either a pre-trained model or a generic all-purpose embedding) and that the model loss is Hölder continuous with respect to these embeddings, we provide a new data selection approach based on $k$-means clustering and sensitivity sampling.

We prove that our new approach allows to select a set of "typical" $k + 1/\varepsilon^2$ elements whose average loss corresponds to the average loss of the whole dataset, up to a multiplicative $(1 \pm \varepsilon)$ factor and an additive $\varepsilon \lambda \Phi_k$, where $\Phi_k$ represents the $k$-means cost for the input data and $\lambda$ is the Hölder constant. Our approach is particularly efficient since it only requires $k$ inferences from the model. We furthermore demonstrate the performance of our approach on classic datasets and show that it outperforms state-of-the-art methods.

We also show that our sampling strategy can be used to define new sampling scores for regression, leading to a new active learning strategy that is comparatively simpler and faster than previous ones like leverage score.

## 1 INTRODUCTION

The rise of both massive data and massive models has led to a new generation of machine learning models with astonishing performances. Yet, the sizes of the models and data make their training extremely difficult, costly, time-consuming, and so nearly impossible for most academic institutions or small-scale companies. On the other hand, a complete dataset is often not needed to reach nearly optimal performances (up to a small increase in error percentage). A central question then becomes how to identify the most important elements for the training process. When dealing with complex, unbalanced data, one may need to depart from uniformly choosing a sample of the data. To achieve this, data selection and active learning methods use the model to iteratively provide (possibly noisy) information about which data elements are the most relevant for training, based on what the model has already learned. There exist several heuristics or greedy approaches for active learning and data selection (see e.g. Dasgupta (2004) or references in Ren et al. (2021)). Successful data selection strategies include *margin* or *entropy* scores with the objective of identifying elements for which the model is the less certain about, and so focusing the training on these elements to improve the performances.

In a celebrated result, Sener & Savarese (2018) showed that active learning strategies are difficult to use in modern training frameworks for the following reasons:

1. The training proceeds in batches, which requires the active learning strategy to not pick *only one* training element at a time but a *batch* of training elements. However, to make the most out of the batch, it is needed to ensure some diversity in the set of elements sampled, which often unfortunately anti-correlates with, for example, the margin objective (see the discussion in Sener & Savarese (2018) for more details).

2. The score (i.e. value) of the training elements are obtained through the model. This requires running the model on the data elements to determine which ones to pick next. Unfortunately, modern models are often very large and the inference time is particularly costly.

The impressively successful approach of Sener & Savarese (2018) consists in using the notion of *coreset* to make the selection. A coreset is a subset of the data defined such that optimizing the model for the coreset yields a good model for the entire data (i.e., good generalization bounds for the whole dataset). In more formal terms, the average loss function for the coreset elements is the same as for the whole dataset and so learning on the coreset elements has the same effect as learning on the whole data. Unfortunately, to implement this approach one would need to obtain the loss of *all* the input elements, which implies running the model on all the input elements. To cirumvent this problem, Sener and Savarese show that, given some embedding representation of the input and some set of assumptions relating the embeddings to the network loss, some form of coreset can be computed using a heuristic to the $k$-center objective. Their embedding assumption is a fairly natural one since the embeddings can be obtained from the network itself (after some basic pre-training) or from a generic embedding model (e.g., BERT Devlin et al. (2018), word2vec Mikolov et al. (2013)).

The above approach has, however, the following suboptimal behaviors:

1. The first practical issue is that the $k$-center objective is particularly sensitive to outliers, and in particular the greedy 2-approximation algorithm in the work of Sener & Savarese (2018) since it iteratively picks the training elements that are the furthest away (in the embedding space) from the already selected training elements. This tends to select outliers. We ask: Can we find a more robust way of selecting a set of elements which is both diverse and that precisely covers the most important traits of the data?

2. A second theoretical drawback is that the bounds proven are quite weak and require strong assumptions on the relationship between the embeddings of the training elements and the model loss, in particular on the spread of the data elements (see Section 2.1 for more details). We ask: Can we provide a theoretical solution that would require a minimal set of assumptions on our data and model?

3. The approach requires to first query a pool of uniformly chosen examples (that is enriched at a later stage). We ask: Can we better select this pool of elements to get the most value out of the sampling mechanism?

4. Finally, and maybe most importantly, their approach is limited to classification tasks. We ask: can we provide a more generic data-selection algorithm, working for a more general loss function?

## 1.1 OUR APPROACH AND CONTRIBUTION

We propose a new data selection algorithm that relies on clustering and sketching methods. Our new approach provides both strong theoretical bounds and significant practical improvements over state-of-the-art data selection methods on classic datasets.

We start from the fact that the $(k, z)$-clustering objective (e.g., $k$-median for $z = 1$ and $k$-means for $z = 2$) provides a more robust clustering measure than $k$-center as it is much less sensitive to outliers. Hence, we start by computing such a clustering, and sample a batch of elements using *sensitivity sampling*, meaning sampling each element with probability proportional to its distance to the closest mean plus the mean's loss (see Feldman & Langberg (2011) for the introduction of that probability distribution for clustering coreset, see also Bachem et al. (2018)) Our experiments indeed show that this yields a better sample in practice than what was previously known, both for neural network and for regression tasks.

Next, we provide theoretical guarantees on this sampling strategy. First, assuming that the model loss is Hölder with respect to the embeddings – a fairly common and well-motivated assumption, more general than the Lipschitz assumption also made by Sener & Savarese (2018) – we can prove that the samples do provide a strong proxy for the loss of all the data elements. More specifically, we show that by using sensitivity sampling we obtain a coreset with respect to the loss of the model, plus an additive term corresponding to the loss of the $(k, z)$-clustering objective. This implies that if the embeddings of the data exhibit some clusterability property, we obtain an actual coreset for

the model loss with only few inferences. Moreover, for classification tasks, expecting that the model embeddings will be clusterable is not an unrealistic assumption: we do expect that points from the same class have closer model-embedding distance than points in different classes. Formally, under the assumption that the loss function $\ell$ is $(z, \lambda)$-Hölder continuous, namely for all $x, y$, $|\ell(x) - \ell(y)| \leq \lambda \|x - y\|^z$, we get:

**Theorem 1.** *[See formal statement in theorem 5] Let $\varepsilon, z, \lambda > 0$. Let $\mathcal{D}$ be a dataset and $\ell$ a loss function that is $(z, \lambda)$-Hölder continuous. Then there exists an algorithm that makes $k$ queries to $\ell$ and outputs a sample $S$ of size $O(\varepsilon^{-2})$ and a weight function $w$ such that*

$$|\sum_{e \in \mathcal{D}} \ell(e) - \sum_{e \in S} w(e)\ell(e)| \leq \varepsilon \left( \sum_{e \in \mathcal{D}} \ell(e) + 2\lambda \Phi_{k,z}(X) \right),$$

*with constant probability where $\Phi_{k,z}(\mathcal{D})$ is the $(k, z)$-clustering cost of $\mathcal{D}$.*

In this theorem, we assume $(z, \lambda)$-Hölder continuity, for some $z$. For instance, when $z = 1$, this encapsulates Lipschitzness and $\Phi_{k,1}(\mathcal{D})$ is the $k$-median cost of $\mathcal{D}$. Therefore, we believe this condition to be quite weak; furthermore, we show that there exist loss functions that are not Hölder continuous for which it is necessary to query the whole dataset in order to get meaningful results. Thus, this answers question 2 raised above.

Except for this assumption, the loss function and the structure of the problem can be quite general: therefore, this answers question 4. We complement this with a similar theorem for linear regression. This completes the answer for question 4.

Second, the $(k, z)$-clustering cost is defined as $\min_{|C|=k} \sum_{e \in \mathcal{D}} \min_{c \in C} \|e - c\|^z$. Instead, the $k$-center guarantee of Sener & Savarese (2018) would translate here into an upper bound $n \cdot \lambda \cdot \min_{|C|=k} \max_{e \in \mathcal{D}} \min_{c \in C} \|e - c\|$. Therefore, depending on the choice of $z$, our upper-bound can be made more or less robust to outliers: the smaller the $z$ the more resilient to outliers it becomes; for large $z$ the objective becomes the $k$-center objective of Sener & Savarese (2018). This addresses question 1 raised above.

Finally, our initial batch of query consists of the $k$ points selected by minimizing the $(k, z)$-clustering cost of $\mathcal{D}$. Instead of a uniform sampling, we therefore select some form of representatives of the dataset, which may give a better representation of it – especially when the dataset admits some clustered structure.

We further demonstrate that the resulting sampling strategy outperforms classic data selection approaches, namely, training the model using the set $S$ obtained via theorem 1 gives a better accuracy than using other methods. Similarly, for linear regression, we show empirically that our sampling strategy is competitive and sometimes outperforms more sophisticated state-of-the-art methods, such as leverage score sampling, adding a fundamentally new sampling strategy to the growing body of work on active regression Chen & Price (2019); Chen & Derezinski (2021); Parulekar et al. (2021); Musco et al. (2022); Woodruff & Yasuda (2023).

We defer a detailed survey of related work to appendix A.1

## 2 PROBLEM FORMULATION

Given a dataset $\mathcal{D}$ and a machine learning model, the high-level goal is to find a subset $S$ of $\mathcal{D}$ such that training the model on $S$ yields approximately the same model as training the model on $\mathcal{D}$, while the time taken to compute $S$ and train the model on $S$ should be much smaller than the time taken to train the model on $\mathcal{D}$.

We focus here on the general data selection problem, and dedicate section 4 to the special case of linear regression.

### 2.1 OUR MODEL

We assume that we are given a dataset $\mathcal{D}$ of size $n$, together with a loss function $\ell$ such that $\ell(e)$ is the loss of the model on instance $e$. The goal is to sample $S \subseteq \mathcal{D}$ of limited size, and associate a

weight function $w : S \mapsto \mathbb{R}_+$ such that

$$\Delta(S) := | \sum_{e \in \mathcal{D}} \ell(e) - \sum_{e \in S} w(e)\ell(e) | \le \delta,$$

for the smallest possible $\delta$. Note that $\ell(e)$ can be queried simply by running the model on $e$ and computing the loss for $e$, so the goal is to compute $S$ without having to compute $\ell(e)$ for all $e \in \mathcal{D}$.

We now provide a complete formulation of the problem.

**Definition 2** (Data Selection, Sener & Savarese (2018)). *The data selection problem is defined as follows:*

- ***Input:*** *A dataset $\mathcal{D}$, an oracle access to a function $\ell : \mathcal{D} \mapsto \mathbb{R}_+$, and a target size $s$.*

- ***Output:*** *A sample $S \subseteq \mathcal{D}$ of size at most $s$ together with a weight function $w : S \mapsto \mathbb{R}_+$ such that*

    - *The number of queries to $\ell$ (i.e.: inferences) is at most $s$.*
    - *$S$ minimizes*

    $$\Delta(S) := | \sum_{e \in \mathcal{D}} \ell(e) - \sum_{e \in S} w(e)\ell(e) |. \tag{1}$$

Note two differences with the the original definition of Sener & Savarese (2018). (A) First, they use uniform weights, namely $\forall s \in S, w(s) = \frac{|\mathcal{D}|}{|S|}$ (in which case, minimizing eq. (1) means that the average $\ell(e)$ in the sample should be close to the average $\ell(e)$ for the whole data). We slightly generalize the definition to allow for different sampling strategies, while keeping an unbiased estimator.

(B) Second, Sener & Savarese (2018) consider the loss after re-training the model with $S$, namely $\left| \frac{1}{n} \sum_{e \in \mathcal{D}} \ell(e, \mathcal{A}(S)) - \sum_{e \in S} w(e)\ell(e, \mathcal{A}(S)) \right|$. In words, the loss of the model trained on $S$ is roughly the same evaluated on $S$ than on $\mathcal{D}$. In order to bound this quantity, Sener and Savarese make *strong assumptions on the distribution of the dataset*, namely the labels are drawn randomly from a structured distribution, and it is further assumed that the training loss is $0$ on their sample. Instead, we stay more general and focus on the loss on the current model. Our underlying assumption is that, if $S$ approximates the loss well, then it contains typical elements of the dataset (with respect to the current model), and therefore updating the model based on $S$ should be similar as if trained on $\mathcal{D}$. This formulation allows us to show strong theoretical results for the data selection problem, *without any assumption on $\mathcal{D}$*. Furthermore, our objective is more challenging than the one from Sener and Savarese: proving a bound on $\Delta(S)$ implies the result of Sener & Savarese (2018) (under their assumption abound the model loss). Therefore, we focus here on $\Delta(S)$.

### 2.2 Assumptions on $\ell$

**Limits to the general case**   The above formulation places us in the sublinear (in $|\mathcal{D}|$) query time regime. Thus, as long as $s = o(|\mathcal{D}|)$ it is impossible to bound $\Delta(S)$ without further assumptions on $\ell$ or $\mathcal{D}$, which can be seen by the following worst-case instance: The adversary chooses uniformly at random (u.a.r.) an element $e^* \in \mathcal{D}$ and define $\ell(e^*) = 1$ and $\ell(e) = 0$ for all $e \neq e^*$. Then computing with constant success probability a sample $S$ of size $s = o(\mathcal{D})$ such that $\Delta(S) = o(\sum_{e \in \mathcal{D}} \ell(e))$ with $o(|\mathcal{D}|)$ queries to $\ell$ is impossible.

**Hölder continuity**   However, in practice we can assume that each element $e$ in $\mathcal{D}$ could be associated with a vector $v_e$ in $\mathbb{R}^d$ for some $d$, possibly coming from the model, that is "well-behaved" with respect to the loss function of the model . More formally, the embeddings of the data elements can either be obtained from a generic embedding of the input dataset $\mathcal{D}$, for example, the BERT or word2vec embeddings for words Devlin et al. (2018); Mikolov et al. (2013), or an embedding obtained through the last layers of the model being trained. The last assumption is particularly realistic in the *warm* start regime where the model has already be partially trained and we are interested in fine-tuning it or training with a sample of the rest of the data.

Our unique assumption is that the loss $\ell$ is $(z, \lambda)$-Hölder continuous with respect to the collection $\{v_e\}_{e \in \mathcal{D}}$, i.e., there exist real-valued constants $z \ge 1, \lambda > 0$ such that for any $e, e' \in \mathcal{D}$

$$|\ell(e) - \ell(e')| \le \lambda ||v_e - v_{e'}||_2^z.$$

We will work in particular with $z = 2$ in our experiments, but our theoretical analysis holds for general $z$ – i.e., when $z = 1$ and the function is Lipschitz. Lipschitzness is a common assumption, and is theoretically grounded for some embeddings (see e.g. Lemma 1 in Sener & Savarese (2018) for CNN). Our assumption relaxes slightly Lipschitzness, and our theoretical finding are therefore more general.

For simplicity, we will assimilate in the following each element $e \in \mathcal{D}$ with its embedding in $\mathbb{R}^d$: we will and denote $e$ the embedding of $e$ (instead of $v_e$).

The problem we consider throughout the rest of the paper is the *Data Selection under $(z, \lambda)$-Hölder Continuity* problem, which is the problem of definition 2 when the loss function $\ell$ is $(z, \lambda)$-Hölder continuous.

This definition can be extended to the active learning setting where the objective is to iteratively choose a set of elements to sample based on the model updates.

**Definition 3** ($r$-Adaptive Active learning under $(z, \lambda)$-Hölder Continuity)**.** *The $r$-adaptive active learning problem under $(z, \lambda)$-Hölder Continuity is defined as follows:*

- ***Input:*** *A set of elements $\mathcal{D} \subset \mathbb{R}^d$, an oracle access to a function $\ell : \mathcal{D} \mapsto \mathbb{R}_+$ that is $(z, \lambda)$-Hölder continous, a target size $s$ and an adaptivity parameter $r$.*

- ***Adaptivity:*** *There are $r$ rounds. At round $i$, the algorithm can query $\ell$ on a set $Q_i$ of size at most $s$. $Q_i$ can only be defined based on the results of $\ell$ on $\cup_{j<i} Q_j$ and $\mathcal{D}$.*

- ***Output:*** *For all $i \in [r]$, a sample $S_i \subseteq \mathcal{D}$ of size at most $s$ together with a weight function $w_i : S \mapsto \mathbb{R}_+$ such that*

  - *For each $i$ the number of queries to $\ell$ is at most $s$.*
  - *$S_i$ minimizes*

$$\Delta(S) := |\sum_{e \in \mathcal{D}} \ell(e) - \sum_{e \in S_i} w_i(e)\ell(e)|.$$

### 2.3 CLUSTERING PRELIMINARIES

We defer a discussion on clustering preliminaries to appendix A.2. Most importantly, the $(k, z)$-clustering cost of $C$ on $\mathcal{D}$ as $\Phi_z(\mathcal{D}, C) := \sum_{x \in \mathcal{D}} \min_{c \in C} \|x - c\|^z$ and $\Phi_{k,z}(\mathcal{D}) := \min_{C \subset \mathbb{R}^d, |C| \leq k} \Phi_z(\mathcal{D}, C)$. For $z = 1$, this objective corresponds to $k$-median, while for $z = 2$ it corresponds to $k$-means.

## 3 ALGORITHMIC RESULTS

We now present a sampling procedure for the active learning problem defined in the previous section.

Our goal is to build a sampling strategy such that $\sum_{s \in S} w(s)\ell(s)$ is an unbiased estimator of $\sum_{e \in \mathcal{D}} \ell(e)$, and show that the estimator is tightly concentrated around its mean.

### 3.1 ALGORITHM AND LOWER BOUND FOR THE NON-ADAPTIVE CASE

We first focus on the context where the algorithm cannot query function $\ell$ at all. In this case, if one only assumes the loss function to be Hölder continuous the error must scale linearly with both the size of the dataset and its diameter, and this can be achieved by a random sample of the data points, as we show in the following theorem. The proof is deferred to appendix B.1

**Theorem 4.** *Let $\varepsilon, \lambda > 0$. There is a constant $c$, a dataset $\mathcal{D}$ and a loss function $\ell$ that is $(z, 1)$-Hölder such that, when $S$ is a uniform sample of size $1/\varepsilon^2$ with weight function $w(e) = n/s$, it holds with constant probability that*

$$\Delta(S) = |\sum_{e \in \mathcal{D}} \ell(e) - \sum_{e \in S} w(e)\ell(e)| \geq c\varepsilon n \sup_{e \in \mathcal{D}} \ell(e).$$

*Furthermore, this lower bound is tight: for all dataset $\mathcal{D}$ and loss function $\ell$, a uniform sample $S$ of size $s = O(1/\varepsilon^2)$ with weights $w(e) = n/s$ satisfies with constant probability $\Delta(S) \leq \varepsilon n \sup_{e \in \mathcal{D}} \ell(e)$.*

### 3.2 ADAPTIVE ALGORITHMS

The lower bound on $\Delta(S)$ in theorem 4 shows is that one must sample more carefully if good guarantees are desired. As explained in the introduction, we present a sampling strategy that queries at most $O(k)$ many points, and reduce the additive error to $\varepsilon\lambda\Phi_k(\mathcal{D})$. This is always better than $\varepsilon n \Delta$, and, in case the embedding of $\mathcal{D}$ has a clustered structured, can be drastically smaller.

#### 3.2.1 1-ROUND ALGORITHM

In this section, we rephrase theorem 1 more precisely and state the 1-round algorithm. The proof of theorem 5 is deferred to appendix B.2

**Theorem 5.** *Let $\varepsilon, \lambda > 0$. Let $(\mathcal{D}, \ell)$ be any input to the 1-adaptive active learning problem under $(z, \lambda)$-Hölder continuity. Then there exists a constant $c$ and an algorithm that makes $k$ queries to $\ell$ and outputs a sample $S$ of size $\lceil \varepsilon^{-2}(2 + 2\varepsilon/3) \rceil$ and a weight function $w$ such that*

$$\Delta(S) = |\sum_{e \in \mathcal{D}} \ell(e) - \sum_{s \in S} w(s)\ell(s)| \leq \varepsilon \left( \sum_{e \in \mathcal{D}} \ell(e) + 2\lambda\Phi_k(X) \right),$$

*with constant probability where $\Phi_k(\mathcal{D})$ is the $(k, z)$-clustering cost of $\mathcal{D}$.*

---

**Algorithm 1** Data-Selection($\mathcal{D}, k, \varepsilon$)

---

1: Compute a $O(1)$-approximation $\mathcal{A}$ to the $(k, z)$-Clustering objective on $\mathcal{D}$, and query $\ell$ on every element of $\mathcal{A}$.
2: For $e \in \mathcal{D}$, define $\mathcal{A}(e) = \arg\min_{a \in \mathcal{A}} \|e - a\|$ the element of $\mathcal{A}$ that is the closest to $e$, $\hat{\ell}(e) := \ell(\mathcal{A}(e))$ and $v(e) := \|e - \mathcal{A}(e)\|^z$.
3: Let $s := \lceil \varepsilon^{-2}(2 + 2\varepsilon/3) \rceil$. For $e \in \mathcal{D}$ define $p_e := \frac{\hat{\ell}(e) + \lambda v(e)}{\lambda\Phi(\mathcal{D}, \mathcal{A}) + \sum_{x \in \mathcal{D}} \hat{\ell}(x)}$ and $w(e) = s^{-1}p_e^{-1}$.
4: Compute a sample $S$ of $s$ points, picked independently following the distribution $p_e$.
5: **Output:** the set $S$ with weights $w$.

---

#### 3.2.2 $r$-ROUND ALGORITHM

We now turn to obtain better guarantees than the above bounds by phrasing the error in terms of the $k$-means loss

**Theorem 6.** *Let $\varepsilon, \lambda, r > 0$. Let $(\mathcal{D}, \ell)$ be any input to the active learning problem under $(z, \lambda)$-Hölder continuity. Then there exists a constant $c$ and an algorithm that for each round $i \in [r]$, queries $k$ elements per round and outputs a sample $S_i$ of size at most $O(1/\varepsilon^2)$ and a weight function $w_i$ such that*

$$\Delta(S_i) = |\sum_{e \in \mathcal{D}} \ell(e) - \sum_{s \in S} w_i(s)\ell(s)| \leq \varepsilon \left( \sum_{e \in \mathcal{D}} \ell(e) + \lambda\Phi_{k \cdot i}(\mathcal{D}) \right)$$

Note that the above algorithm allows to trade-off the round complexity and sample size and reaches optimality in the limit: when $r \cdot k = |\mathcal{D}|$, we obtain an exact algorithm. The algorithm is very similar to algorithm 1: We defer the presentation to appendix B.3.

## 4 DATA SELECTION FOR REGRESSION

In this section, we specialize our method – sampling according to $(k, z)$-clustering cost – to the setting of linear regression. Ideally, given a matrix $A$, our goal is to compute a sketching and rescaling diagonal matrix $S$ with as few as possible non-zero entries such that computing the optimal

regression on $S$ is equivalent to computing it on $A$. For this, we are seeking a coreset guarantee, namely we want $\|SAx - b\| \approx \|Ax - b\|$, for all $x$. In the following $a_i$ denotes the $i$-th row of $A$.

Therefore, we define the data selection problem for regression as follows:

**Definition 7.** *The data selection problem for linear regression is defined as follows:*

- ***Input:*** *a $n \times d$ matrix $A$ and a $n$-dimensional vector $b$, and a target number of queries $k$.*

- ***Output:*** *A sample $S \subseteq [n]$ of size at most $s$ together with a weight function $w : S \rightarrow \mathbb{R}_+$ such that $\left| \sum_{s \in S} w(s)(\langle a_s, x \rangle - b_s)^2 - \|Ax - b\|_2^2 \right|$ is as small as possible, for all $x$.*

We do not achieve such a general statement, and need to make several assumptions, both on $A, b$ and to restrict the set of possible $x$. Our first set of assumptions is the following:

**Assumption 8.** *For all $i, \|a_i\|_2 = O(1)$ and $b_i = O(1)$. Furthermore, there is a constant $\zeta$ such that for all $i, j$, $|b_i - b_j| \leq \zeta \|a_i - a_j\|_2$.*

The above assumption is similar to the Lipschitz (or more generally Hölder) assumption we made for theorem 1: it formalizes that the labels (i.e., $b_i$s) must be close when the embeddings (i.e., the $a_i$s) are close. As before, it necessary to get any result querying a sublinear number of labels $b_i$. This assumption is also related to that of Sener & Savarese (2018) who assume the labels of the data are drawn randomly, following distributions that are Lipschitz.

The basic idea of our algorithm is to interpret each row of $A$ as a point in $\mathbb{R}^d$ and cluster these points using $k$-median. Then we compute the optimal regression $x_0$ for the dataset consisting of all centers, each weighted by the size of its cluster, and use $x_0$ to define a probability distribution over all points. Sampling $s$ points according to this distribution gives a set $S$ together with a suitable weight function

---

**Algorithm 2** Data-Selection-Regression($A, k, \varepsilon, \zeta$)

1: Compute a $O(1)$-approximation $\mathcal{A}$ to $k$-median on $A$.
2: For all $i \in [n]$, let $j$ be such that $a_j$ is the closest center to $a_i$ in $\mathcal{A}$: define $\hat{a}_i = a_j$, $\hat{b}_i = b_j$ and $v(a_i, x_0) = (\langle \hat{a}_i, x_0 \rangle - \hat{b}_i)^2$.
3: Compute the optimal regression $x_0$ for the dataset $\{\hat{a}_1, ..., \hat{a}_n\}$, i.e., the dataset where each center of $\mathcal{A}$ is weighted by the size of its cluster.
4: Define $p_i := \frac{\zeta \|a_i - \hat{a}_i\| + v(a_i, x_0)}{\sum_{j \in [n]} \zeta \|a_j - \mathcal{A}(a_j)\| + v(a_j, x_0)}$, and $w(i) = s^{-1} p_i^{-1}$.
5: Compute a sample $S$ of $s$ points, picked independently following the distribution $p_i$.
6: **Output:** the set $S$ with weights $w$.

---

Our main theorem for regression is stated next. The proof is deferred to appendix B.4.

**Theorem 9.** *Let $\zeta \geq 1$, $A$ and $b$ respecting assumption 8 with $\zeta$ being constant, and let $\hat{a}_i$ and $x_0$ as computed by algorithm 2.*

*Let $\mathcal{X}$ be the set of vectors $x$ such that $\|x\|_2 = O(1)$ and $\forall i, |\langle \hat{a}_i, x - x_0 \rangle| \leq \zeta \|a_i - \hat{a}_i\|_2$.*

*For $s = O(d/\varepsilon^2 \log(1/\delta))$, it holds with probability $1 - \delta$ that, for all $x \in \mathcal{X}$,*

$$\left| \sum_{s \in S} w(s)(\langle a_s, x \rangle - b_s)^2 - \|Ax - b\|_2^2 \right| \leq \varepsilon(\|Ax - b\|_2^2 + \zeta \Phi_{k,1}(A)))$$

## 5 EXPERIMENTS

We consider the setting of data selection, in which the goal is to train a model on a new dataset in a runtime-efficient and data-efficient way. Runtime efficiency means that we prefer an algorithm with low total runtime and a *sublinear* number of queries to a model inference oracle, and data efficiency means that there is a bound on the number of data points that can be used for training because labels are costly or hard to obtain.

In Section 5.1 we present experiments on a linear regression task, and in Section 5.2 we present results on some common neural network benchmark tasks.

## 5.1 EXPERIMENTS ON LINEAR REGRESSION

Following our theoretical analysis in Section 4, we validate our coreset sampling algorithm on a linear regression task. We present our results on the UCI gas sensor dataset[1] Vergara et al. (2012); Rodriguez-Lujan et al. (2014) in Figure 1.

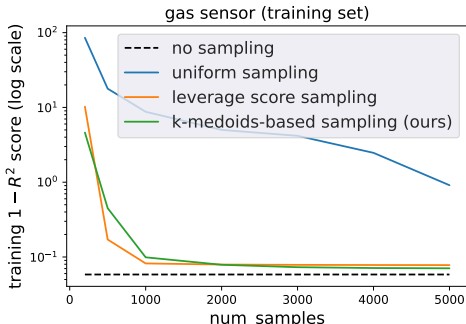
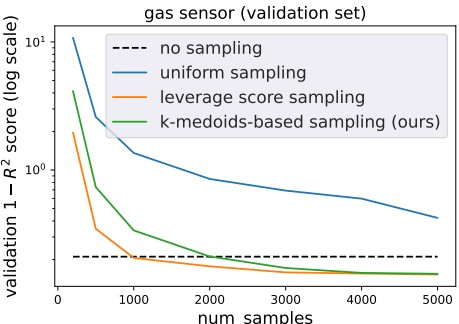

Figure 1: Experimental results on the gas sensor regression dataset. We plot the $1 - R^2$ error on the training and validation set (lower is better). To minimize variation, we independently run each data point 100 times and average the results.

**Our algorithm.** We run algorithm 2, with a couple of differences: i) We run $k$-medoids, which is a variant of $k$-median, and ii) we set $\zeta \to \infty$, which has the effect that we only look at distances and not losses. We set the number of clusters to be $10\%$ of the total number of data points in the training set. After computing the clustering, we randomly sample each data point with probability $p_e$ proportional to the distance from the nearest cluster center, and re-weigh each sampled term in the regression objective by $1/p_e$. After computing the regression solution, we evaluate it on the full training and validation datasets.

We compare with uniform sampling and leverage score sampling. Leverage score sampling is the de facto sampling algorithm for linear regression, and it is known to have extremely good performance but high runtime cost, since it requires solving a full-dimensional linear system per example data point. Surprisingly, we find that our clustering-based algorithm performs almost on par with leverage score sampling, while being drastically faster – the $k$-medoids solution can be computed in linear time.

## 5.2 EXPERIMENTS ON NEURAL NETWORKS

For our neural network experiments, our setting is as follows: Given a target number of data points $k$ that we need to sample, we first train an initial model using a uniformly random subset of $k' < k$ data points, and then run a model-based sampling algorithm that uses this initial model to sample the remaining $k - k'$ data points. Finally, we train a model on all the $k$ data points and evaluate it on a held-out validation set. For our experiments, we chose $k' = 0.2k$.

**Our algorithms.** We consider two instantiations of the sensitivity sampling algorithm presented in Algorithm 1: *loss-based sampling* and *gradient-based sampling*.

- Loss-based sampling ("loss"): we set $\ell(e) := L(y, \text{model}_\theta(e))$ to be the *loss* of the model on example $e$ with respect to the true label $y$, where $\theta$ are the model parameters.

- Gradient-based sampling ("grad"): we set $\ell(e) := \|\nabla_\theta L(y, \text{model}_\theta(e))\|_2^2$ to be equal to the squared $\ell_2$ norm of the gradient update.

For the clustering required in Algorithm 1, we run $k''$-means clustering using python's sklearn implementation, for some $k'' < k$ on the model's last layer embeddings. For our experiments, we

---

[1]`https://archive.ics.uci.edu/ml/datasets/Gas+Sensor+Array+Drift+Dataset+at+Different+Concentrations`

chose $k'' = 0.2k$. Since the $k''$ cluster centers might not be actual data points from the dataset, we replace each center with the closest data point from the dataset in $\ell_2$ norm (note that by triangle inequalities, this loses only a factor 4 in the $k$-means cost). After computing $\ell(e)$ for each center and extrapolating to the whole dataset using the approximation $\widetilde{\ell}(e) := \ell(\mathcal{A}(e)) + \lambda \|e - \mathcal{A}(e)\|_2^2$, we sample the remaining $k - k' - k''$ data points proportional to $\widetilde{\ell}$.

We present our results in Figure 2, where we compare with two algorithms: i) uniform sampling ("uniform"), which samples a subset of examples uniformly at random, which is the simplest possible baseline for data selection, and ii) the $k$-center coreset algorithm of Sener & Savarese (2018) ("[SS18]"). We notice that the loss- and gradient-based sampling algorithms from algorithm 1 perform best when the number of samples is relatively small. In addition, based on the runtime comparison in figures 2 and 3, the loss-based algorithm performs best in terms of runtime.

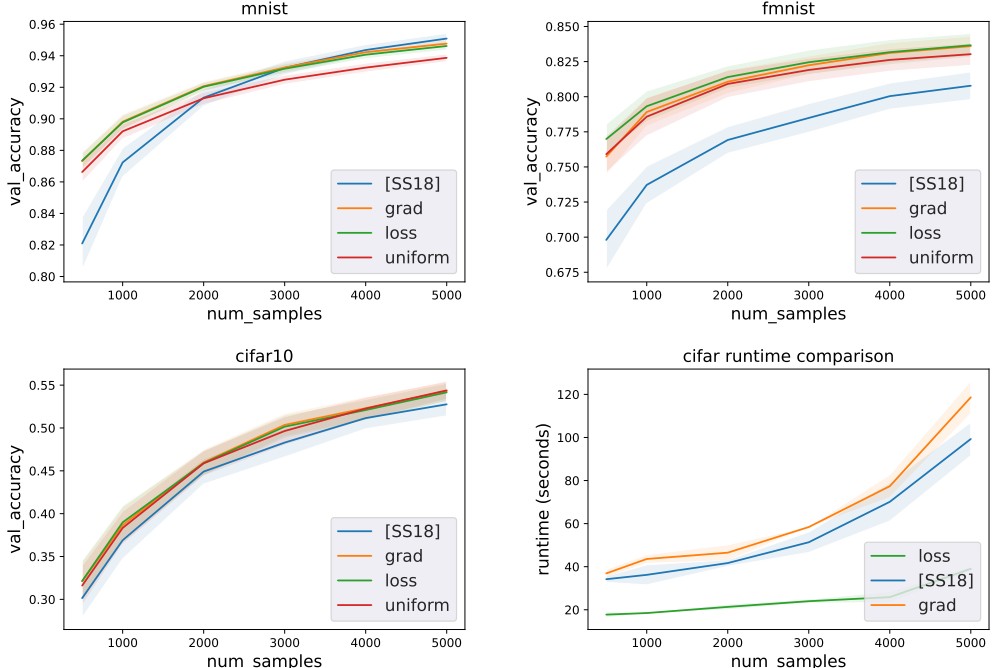

(a) Plots of experimental results for different datasets. For each algorithm, we plot the accuracy on the validation dataset for different values of $k$ (number of samples). We also provide a runtime comparison on CIFAR10.

| Algorithm | MNIST | Fashion MNIST | CIFAR10 |
|---|---|---|---|
| uniform | 0.9130 | 0.8091 | 0.4587 |
| coreset Sener & Savarese (2018) | 0.9134 | 0.7692 | 0.4491 |
| Loss-based Algorithm 1 | 0.9203 | **0.8140** | 0.4590 |
| Gradient-based Algorithm 1 | **0.9207** | 0.8107 | **0.4598** |

(b) Experimental results for $k = 2000$ and different datasets. For each algorithm, we show the accuracy on the validation dataset.

Figure 2: The experimental comparison. To minimize variation, we independently run each data point 100 times, and present the mean with bands of one standard deviation.

# 6 CONCLUSIONS

We presented a fast, high-quality algorithm for selecting a training set. We showed that it has strong theoretical bounds and also performs very well in our empirical evaluation comparing it to slower, state-of-the-art active learning algorithms.

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

## A   PRELIMINARIES

### A.1   FURTHER RELATED WORK

We present related work in data selection and active learning. For extensive and recent survey, we refer to Ren et al. (2021), and detail here some of the most relevant point for comparison.

Our work departs from previous work in the following ways: We work in the regime where we have a budget on both the number of elements selected (i.e.: labeled) and the number of inferences of the model. Moreover, we present a rigorous analysis of our sampling mechanism that works for a variety of machine learning models, as long as we are provided with embeddings that are Lipshitz with respect to the loss of the model.

Given an unlabeled set of points, the active learning problem asks to identify the most relevant points to label (Settles (2009); Cohn et al. (1996)). In the era of big data, labeling a big dataset is often too expensive. We are thus given a budget of, say, $k$ elements that we can label. The question becomes how to pick these $k$ elements so as to maximize the performance of the final model (that will be trained on these elements).

From a theoretical standpoint, Dasgupta (2004) demonstrated that greedy active learning heuristics perform poorly if agnostic to both data and learning algorithm. To circumvent these negative results, other works have made assumptions on the data-dependent realizability of the hypothesis space like (Gonen et al. (2013)) or on a data dependent measure of the concept space called disagreement coefficient (Hanneke (2007)).

Related to ours, several successful works have brought together unsupervised techniques such as clustering and information from the model (such as margin scores), see e.g.: Citovsky et al. (2021). The work of Sener & Savarese (2018) brings together clustering, and sketching techniques (coresets in this case).

Another line of works consists of bayesian active learning methods which use a non-parametric model, like a Gaussian process, to obtain an estimate of the expected improvement on the model after each query (Kapoor et al. (2007)), or alternatively the expected error after a set of queries (Roy & McCallum (2001)). It seems that an important drawback is that these approaches do not scale to large models (see the discussion in Sener & Savarese (2018)).

Uncertainty based methods form another important family of active learning algorithm. They aim at finding relevant examples using heuristics like highest entropy (Joshi et al. (2009)), or geometric distance to decision boundaries (Tong & Koller (2001); Brinker (2003)).

Batch-active learning based on uncertainty may lead to a useless batch, where all queries are very similar (when the highest uncertainty is concentrated in a small region). To cope with this, several methods that aim at trading-off diversity and uncertainty to select the points. The way the elements are iteratively selected can vary depending on the application from mini-batches to one-shot (e.g.: Hoi et al. (2006); Guo & Schuurmans (2007); Chakraborty et al. (2014); Citovsky et al. (2021); Amin et al. (2020)) Specifically in the context of mini-batch active learning, a common approach is to use unsupervised machine learning techniques to extract information from the data. Such methods include $k$- Medoid (Schubert & Rousseeuw (2019)), or MaxCover (Hochbaum & Pathria (1998)) to select a set of data points that maximally cover the dataset with respect to some objective. Elhamifar et al. (2013) and Yang et al. (2015) design a discrete optimization problem for this purpose, that they solve using convex optimization methods. Unfortunately, the running time of these methods is quadratic in the input data size and so highly impractical for large data.

Covering or clustering approaches have also been tried in the past Joshi et al. (2010); Wang & Ye (2015). The former does not provide any theoretical guarantee associated to its approach. The latter uses empirical risk minimization to minimize the difference between the maximum mean discrepancy between iid. samples from the dataset and the actively selected samples (instead of the loss we work with).

Active learning has also been studied when tailored to some specific machine learning problems such as nearest neighbors, logistic regression or linear regression with Gaussian noise (Wei et al. (2015); Hoi et al. (2006); Guo & Schuurmans (2007); Yu et al. (2006)).

Recently, active learning was also extended to $\ell_p$-regression for all $p \geq 1$ without any assumptions on the data, resulting in a number of optimal bounds Chen & Price (2019); Chen & Derezinski (2021); Parulekar et al. (2021); Musco et al. (2022); Woodruff & Yasuda (2023). These works are based on using sampling probabilities defined from the design matrix (agnostic to the label vector), and range from leverage scores ($p = 2$) to $\ell_p$-sensitivities to $\ell_p$-Lewis weights, the latter achieving optimal bounds for all $p \geq 1$. Our work adds a new set of scores to this growing literature for regression, namely, scores that are proportional to the cost of clustering individual points. We note that in practice, computing an approximate $k$-median solution may be much faster than approximating the Lewis weights or leverage scores of a matrix since it does not involve computing the inverse of any matrices.

If the model can be run on all input data, and so the confidence of the model is known for all the input data elements, then several set-cover based methods that aims at best covering the hypothesis space have been designed (Guillory & Bilmes (2010); Golovin & Krause (2011); Esfandiari et al. (2021). The key distinguishing factor of our approach compared to these works is that we do not require the model to be run on all the input data. Furthermore, as we demonstrate, our sampling technique applies more generally to problems other than regression.

**Coresets** Our ideas are inspired from the coreset literature. Coreset were introduced initially for $k$-median and $k$-means clustering: the goal is to compute a (weighted) set $S$ of points such that, for any set of $k$ centers, evaluating its cost on $S$ is almost the same as evaluating it on the full dataset Har-Peled & Mazumdar (2004). Coresets with optimal size (which is $O(k\varepsilon^{-2}\min(\varepsilon^{-2}, \sqrt{k}))$) exist Cohen-Addad et al. (2021; 2022); Huang et al. (2023), and one of the most standard tool to build a coreset is sensitivity sampling, namely sampling according to the cost in a constant-factor $(k, z)$-clustering solution.

Ideas from the literature on coreset for clustering have already spread to other domains: Tukan et al. (2023) presents coresets for Radial basis function neural networks of small sizes, and Tukan et al. (2020b) for near-convex functions. Mussay et al. (2020); Tukan et al. (2022) showed how to use coreset for prunning and compressiong neural networks, and Maalouf et al. (2022) used coreset for fast least-square linear regression. For machine-learning tasks, Tukan et al. (2020a) present coresets for Support Vector Machines.

A.2 CLUSTERING PRELIMINARIES

In the following, we are given a set of points $\mathcal{D}$ in the Euclidean Space $\mathbb{R}^d$ with $\ell_2$ norm. We let $\mu_z(\mathcal{D})$ be the power mean of $X$, namely the point $p$ that minimizes $\sum_{x \in \mathcal{D}} \|x - p\|^z$. We let $\mathrm{Disp}_z(\mathcal{D}) := \sum_{x \in X} \|x - \mu(\mathcal{D})\|^z$.

Given a set of $k$ points $C \in (\mathbb{R}^d)^k$, we denote the $(k, z)$-clustering cost of $C$ on $\mathcal{D}$ as $\Phi_z(\mathcal{D}, C) := \sum_{x \in \mathcal{D}} \min_{c \in C} \|x - c\|^z$ and $\Phi_{k,z}(\mathcal{D}) := \min_{C \subset \mathbb{R}^d, |C| \leq k} \Phi_z(\mathcal{D}, C)$. For $z = 1$, this objective corresponds to $k$-median, while $k$-means is for $z = 2$. Note that $\mathrm{Disp}_z(\mathcal{D}) = \Phi_{1,z}(\mathcal{D})$.

We say that a set $C$ of $k$ points is an $\alpha$-approximation to $(k, z)$-clustering on $\mathcal{D}$ when $\Phi_z(C, \mathcal{D}) \leq \alpha \Phi_{k,z}(\mathcal{D})$. An ordered list of centers $c_1, ..., c_n$ is an $\alpha$-approximation to Prefix-$z$-clustering when, for all $1 \leq k \leq n$, $(c_1, ..., c_k)$ is an $\alpha$-approximation to $(k, z)$-clustering on $\mathcal{D}$. An $O(1)$-approximation to prefix $(k, z)$-clustering can be computed using the algorithm of Mettu & Plaxton (2003). $D^z$-sampling (which is $k$-means++ for $z = 2$) gives an $O(\log k)$-approximation, which is fast and performs extremely well in practice.

B DEFERRED PROOFS

In this section, we use Bernstein's concentration inequality:

**Theorem 10** (Bernstein's inequality). *Let $X_1, \ldots, X_n$ be independent random variables and let $M > 0$ be such that, for all $i$, $|X_i| \leq M$. Then, for all $t > 0$,*

$$Pr[|\sum_i X_i - E[\sum_i X_i]| \geq t] \leq \exp(-\frac{t^2}{2\sum_{x \in X} E[X_x^2] + 2Mt/3})$$

### B.1 PROOF FOR THE NON-ADAPTIVE CASE

*Proof of theorem 4.* We denote for simplicity $R = \sup_{e \in \mathcal{D}} \ell(e)$. The upper-bound is a simple application of Bernstein's inequality and is included for completeness.

The algorithm chooses successively $s$ uniformly random samples $S_1, ..., S_s$ from $\mathcal{D}$ (with replacement) and gives each sampled element weight $n/s$. Let $X_i = w(S_i)\ell(S_i)$. It holds that $\mathbb{E}[X_i] = \frac{n}{s} \sum_{e \in \mathcal{D}} \frac{\ell(e)}{n} = \frac{\sum_{e \in \mathcal{D}} \ell(e)}{s}$, and, thus, $\mathbb{E}[\sum_{e \in S} w(e)\ell(e)] = \mathbb{E}[\sum X_i] = \sum_{e \in \mathcal{D}} \ell(e)$.

To show that this sum of random variables is concentrated, we aim at applying Bernstein's inequality. For this, we need to bound $\mathbb{E}[X_i^2]$: we have

$$\mathbb{E}[X_i^2] = \sum_{e \in \mathcal{D}} \left(\frac{n}{s}\ell(e)\right)^2 \Pr[e = S_i] \leq \sum_{e \in \mathcal{D}} \frac{n}{s^2}\ell(e)^2 \leq \frac{n^2}{s^2}R^2.$$

Summed over all $i$, we therefore have $\sum_{i=1}^{s} \mathbb{E}[X_i^2] \leq n^2 R^2/s$. Furthermore, for any $i$, $|X_i| \leq \frac{n}{s}R$ with probability 1. Plugging this result into Bernstein's inequality yields

$$\Pr[\Delta(S) \geq \varepsilon n R] = \Pr\left[\left|\sum_i X_i - E[\sum_i X_i]\right| \geq \varepsilon n R\right] \leq \exp\left(-\frac{\varepsilon^2 n^2 R^2 \cdot s}{2n^2 R^2 + 2nR \cdot \varepsilon n R/3}\right)$$

$$\leq \exp(-\varepsilon^2 \cdot s/(2 + \varepsilon)).$$

With $s = O(1/\varepsilon^2)$, this gives the desired probability bound.

For the lower bound, consider a multiset $\mathcal{D} \subset \mathbb{R}$ with $n/2$ copies of $-1$ and $n/2$ copies of 1, and $\ell$ being the identify function $\ell(x) = x$, implying that $\sum_{e \in \mathcal{D}} \ell(e) = 0$. Then, the estimator is a sum of Rademacher random variables, and anti-concentration bound states that for any fixed value $x$, $\sum_{s \in S} \ell(s) = x$ with probability at most $O(1/\sqrt{|S|})$ Littlewood & Offord (1939). Therefore, for some constant $c$ (which depends on the previous big-O constant), $\Pr[|\sum_{s \in S} \ell(s)| \geq c\sqrt{|S|}] \geq 1/2$. Multiplying with $n/|S|$, this implies that our estimator is bigger than $\frac{cn}{\sqrt{|S|}}$ with probability at least 1/2. When $|S| \leq 1/\varepsilon^2$, this gives the desired statement. $\square$

### B.2 PROOF OF THEOREM 5

*Proof.* We prove that algorithm 1 yields the desired result. Let $S_1, ..., S_s$ be the successive random samples, and define $X_i = \ell(S_i)w(S_i)$. Note that $\Phi_z(\mathcal{D}, \mathcal{A}) = \sum_{e \in \mathcal{D}} v(e)$.

Using the Hölder continuity of $\ell$, we get that $\ell(e) \leq \hat{\ell}(e) + \lambda v(e)$ and $\hat{\ell}(e) \leq \ell(e) + \lambda v(e)$. Therefore,

$$
\begin{aligned}
E[X_i^2] &= \sum_{e \in \mathcal{D}} (\ell(e)w(e))^2 \cdot p_e = \sum_{e \in \mathcal{D}} \ell(e)^2 \frac{1}{p_e s^2} \\
&= \frac{1}{s^2} \cdot \sum_{e \in \mathcal{D}} \ell(e)^2 \cdot \frac{\lambda \Phi_z(\mathcal{D}, \mathcal{A}) + \sum_{x \in \mathcal{D}} \hat{\ell}(x)}{\hat{\ell}(e) + \lambda v(e)} \\
&\leq \frac{1}{s^2} \cdot \sum_{e \in \mathcal{D}} \ell(e) \cdot (\hat{\ell}(e) + \lambda v(e)) \cdot \frac{\lambda \Phi_z(\mathcal{D}, \mathcal{A}) + \sum_{x \in \mathcal{D}} \hat{\ell}(x)}{\hat{\ell}(e) + \lambda v(e)} \\
&= \frac{1}{s^2} \cdot \sum_{e \in \mathcal{D}} \ell(e) \cdot \left(\lambda \Phi_z(\mathcal{D}, \mathcal{A}) + \sum_{x \in \mathcal{D}} \hat{\ell}(x)\right) \\
&\leq \frac{1}{s^2} \cdot \left(\sum_{e \in \mathcal{D}} \ell(e)\right) \cdot \left(2\lambda \Phi_z(\mathcal{D}, \mathcal{A}) + \sum_{x \in \mathcal{D}} \ell(x)\right) \\
&\leq \frac{1}{s^2} \cdot \left(2\lambda \Phi_z(\mathcal{D}, \mathcal{A}) + \sum_{e \in \mathcal{D}} \ell(e)\right)^2.
\end{aligned}
$$

We let $M := \max_{e \in \mathcal{D}} \ell(e)w(e) = \max_e \ell(e) \cdot \frac{\lambda \Phi_z(\mathcal{D}, \mathcal{A}) + \sum_x \hat{\ell}(x)}{s(\hat{\ell}(e) + \lambda v(e))}$. As previously, we have $M \leq \frac{1}{s}\left(\lambda \Phi_z(\mathcal{D}, \mathcal{A}) + \sum_{e \in \mathcal{D}} \hat{\ell}(e)\right) \leq \frac{1}{s} \cdot \left(2\lambda \Phi_z(\mathcal{D}, \mathcal{A}) + \sum_{e \in \mathcal{D}} \ell(e)\right)$.

Now, the Bernstein inequality (see theorem 10) implies, for $t = 2\lambda \Phi_z(\mathcal{D}, \mathcal{A}) + \sum_{e \in \mathcal{D}} \ell(e)$:

$$Pr\left[\Delta(S) > \varepsilon t\right]$$
$$\leq \exp\left(\frac{-\varepsilon^2 t^2}{2\sum_{i=1}^s \mathbb{E}[X_i^2] + 2M\varepsilon t/3}\right)$$

Using $M \leq t/s$ and $\mathbb{E}[X_i]^2 \leq \frac{1}{s^2} \cdot t^2$, we get:

$$\Pr\left[\Delta(S) > \varepsilon t\right] \leq \exp\left(-\varepsilon^2 \frac{t^2 \cdot s}{2t^2 + 2\varepsilon t^2/3}\right) = \exp\left(-\varepsilon^2 \frac{s}{2 + 2\varepsilon/3}\right) \leq \exp(-1),$$

where the last inequality holds by the choice of $s = \lceil \varepsilon^{-2}(2 + 2\varepsilon/3) \rceil$. $\qquad \square$

### B.3 ADAPTIVE ACTIVE LEARNING

We present here the algorithm used to prove theorem 6. The proof follows directly from the proof for theorem 5.

---

**Algorithm 3** Adaptive-active-learning($\mathcal{D}, r, \lambda, \varepsilon$)

1: Compute a $O(1)$-approximation to prefix-$z$-clustering on $\mathcal{D}$, i.e., an ordering of the points in $\mathcal{D}$ such that any prefix of length $k_0$ is an $O(1)$-approximation to $(k_0, z)$-clustering on $\mathcal{D}$.
2: **for** each round $i = 1, ..., r$ **do**
3:      query $\ell$ for the points at position in $[(i-1)k + 1, ik]$ in the ordering, and define $\mathcal{A}$ to be the set of elements at position at most $ik$ in the ordering.
4:      For $e \in \mathcal{D}$, define $\mathcal{A}(e) = \arg\min_{a \in \mathcal{A}} \|e - a\|$ the element of $\mathcal{A}$ that is the closest to $x$, $\hat{\ell}(e) := \ell(\mathcal{A}(e))$ and $v(e) := \|e - \mathcal{A}(x)\|$.
5:      Define $p_e := \frac{\hat{\ell}(e) + \lambda v(e)}{\lambda \Phi_z(\mathcal{D}, \mathcal{A}) + \sum_{x \in \mathcal{D}} \hat{\ell}(x)}$.
6:      Let $s := \lceil \varepsilon^{-2}(2 + 2\varepsilon/3) \rceil$. For $e \in \mathcal{D}$ define $p_e := \frac{\hat{\ell}(e) + \lambda v(e)}{\lambda \Phi_z(\mathcal{D}, \mathcal{A}) + \sum_{x \in \mathcal{D}} \hat{\ell}(x)}$ and $w_i(e) = s^{-1} p_e^{-1}$.
7:      Compute a sample $S_i$ of $s$ points, picked independently following the distribution $p_e$.
8: **end for**
9: **Output:** $S_i, w_i$ for each round $i$.

---

### B.4 REGRESSION

To show theorem 9, we first prove that for any fixed $x \in \mathcal{X}$, the desired bound hold with probability $1 - \exp(-\varepsilon^2 s)$. It is standard to extend this result to all $x \in \mathcal{X}$, using discretization techniques to find a set $N$ of size $\varepsilon^{-O(d)}$ such that preserving the cost for all vectors in $N$ is enough to extend the result for all candidate $x$ ($N$ is called a net for $\mathcal{X}$). Hence, it is enough to show the following lemma:

**Lemma 11.** *Let $\zeta, \kappa \geq 1$, and $A$ and $b$ that respects assumption 8 with constant $\zeta$ and $\hat{a}_i$ and $x_0$ as computed by algorithm 2.*

*Let $x \in \mathcal{X}$, namely $x$ such that $\|x\|_2 = O(1)$ and there is some $\zeta \geq 1$ such that $\forall i, |\langle \hat{a}_i, x - x_0 \rangle| \leq \kappa \|a_i - \hat{a}_i\|_2$. Then, with probability $1 - \delta$, it holds that for $s = 8\varepsilon^{-2} \log(1/\delta)$,*

$$\left| \sum_{s \in S} w(s)(\langle a_s, x \rangle - b_s)^2 - \|Ax - b\|_2^2 \right| \leq \varepsilon(\|Ax - b\|_2^2 + \zeta \Phi_{k,1}(A)))$$

*Proof.* Let $S_1, ..., S_s$ be the successive random samples, and define $X_t = w(S_t)(\langle a_{S_t}, x \rangle - b_{S_t})^2$. By choice of $w$, it holds that $\mathbb{E}[\sum X_t] = \|Ax - b\|_2^2$. We will show concentration using the Bernstein inequality.

We first focus on bounding the second moment of $X_t$. We have:

$$\mathbb{E}[X_t^2] = \sum_{i=1}^{n} \frac{(\langle a_i, x \rangle - b_i)^4}{s^2 p_i}$$
$$= \frac{1}{s^2} \cdot \sum_{i=1}^{n} (\langle a_i, x \rangle - b_i)^4 \cdot \frac{\sum_{j \in [n]} \zeta \|a_j - \mathcal{A}(a_j)\| + v(a_j, x_0)}{\zeta \|a_i - \hat{a}_i\| + v(a_i, x_0)}$$

To bound this term, we first note that

$$(\langle a_i, x \rangle - b_i)^2 - (\langle \hat{a}_i, x \rangle - \hat{b}_i)^2$$
$$= (\langle a_i + \hat{a}_i, x \rangle - b_i - \hat{b}_i)(\langle a_i - \hat{a}_i, x \rangle - b_i + \hat{b}_i)$$
$$= \langle a_i + \hat{a}_i, x \rangle \langle a_i - \hat{a}_i, x \rangle + \langle a_i + \hat{a}_i, x \rangle \cdot (\hat{b}_i - b_i) - (b_i + \hat{b}_i)\langle a_i - \hat{a}_i, x \rangle - (\hat{b}_i + b_i)(\hat{b}_i - b_i)$$
$$\leq \|a_i + \hat{a}_i\| \cdot \|a_i - \hat{a}_i\| \cdot \|x\|^2 + \|a_i + \hat{a}_i\| \cdot \|x\| |\hat{b}_i - b_i| - |b_i + \hat{b}_i| \cdot \|a_i - \hat{a}_i\| \cdot \|x\| + |b_i + \hat{b}_i| \cdot |b_i - \hat{b}_i|$$
$$= O(\zeta \|a_i - \hat{a}_i\|), \tag{2}$$

where the last two lines follow from Cauchy-Schwarz and assumption 8.

We now relate this to the term $v(a_i, x_0) = (\langle \hat{a}_i, x_0 \rangle - \hat{b}_i)^2$ of the denominator:

$$(\langle \hat{a}_i, x \rangle - \hat{b}_i)^2 - (\langle \hat{a}_i, x_0 \rangle - \hat{b}_i)^2 = \langle \hat{a}_i, x - x_0 \rangle \cdot (\hat{a}_i, x - x_0) + 2\hat{b}_i)$$
$$= \langle \hat{a}_i, x + x_0 \rangle \langle \hat{a}_i, x - x_0 \rangle - 2\hat{b}_i \langle \hat{a}_i, x - x_0 \rangle$$
$$= O(|\langle \hat{a}_i, x - x_0 \rangle|) = O(\kappa \|a_i - \hat{a}_i\|),$$

where the last line uses our assumption $|\langle \hat{a}_i, x - x_0 \rangle| \leq \kappa \|a_i - \hat{a}_i\|$. Thus, combining those equations:

$$(\langle a_i, x \rangle - b_i)^2 \leq O(\zeta \|a_i - \hat{a}_i\|) + (\langle \hat{a}_i, x_0 \rangle - \hat{b}_i)^2 + O(\kappa \|a_i - \hat{a}_i\|)$$
$$= O((\zeta + \kappa)\|a_i - \hat{a}_i\| + v(a_i, x_0)).$$

Thus, we can now finish our bound on the second moment of $X_t$:

$$\mathbb{E}[X_t^2] = \frac{1}{s^2} \cdot \sum_{i=1}^{n} (\langle a_i, x \rangle - b_i)^4 \cdot \frac{\sum_{j \in [n]} \zeta \|a_j - \mathcal{A}(a_j)\| + v(a_j, x_0)}{\zeta \|a_i - \hat{a}_i\| + v(a_i, x_0)}$$
$$\leq \frac{1}{s^2} \cdot \sum_{i=1}^{n} (\langle a_i, x \rangle - b_i)^2 \cdot O((\zeta + \kappa)\|a_i - \hat{a}_i\| + v(a_i, x_0)) \cdot \frac{\sum_{j \in [n]} \zeta \|a_j - \mathcal{A}(a_j)\| + v(a_j)}{\zeta \|a_i - \hat{a}_i\| + v(a_i, x_0)}$$
$$\leq \frac{\sum_{j \in [n]} \zeta \|a_j - \mathcal{A}(a_j)\| + v(a_j, x_0)}{s^2} \cdot \sum_{i=1}^{n} O\left((\langle a_i, x \rangle - b_i)^2\right)$$

Using the same upper bounds, we get that for all $i$,

$$w(i)(\langle a_i, x \rangle - b_i)^2 = \frac{1}{s} \cdot (\langle a_i, x \rangle - b_i)^2 \cdot \frac{\sum_{j \in [n]} \zeta \|a_j - \mathcal{A}(a_j)\| + v(a_j, x_0)}{\zeta \|a_i - \hat{a}_i\| + v(a_i, x_0)}$$
$$\leq \frac{\sum_{j \in [n]} \zeta \|a_j - \mathcal{A}(a_j)\| + v(a_j, x_0)}{s}.$$

Therefore, for $T = \varepsilon(\|Ax - b\|_2^2 + \zeta \Phi_{k,1}(A) + \sum_i v(a_i, x_0))$ we get that $2 \sum_t \mathbb{E}[X_t^2] \leq T^2/s$, and with probability 1 each $X_t$ is verifies $2|X_t|T/3 \leq T^2/s$. Furthermore, using eq. (2) and the

optimality of $x_0$ for the dataset $\{\hat{a}_1, ..., \hat{a}_n\}$, we have that:

$$\sum_i v(a_i, x_0) = \sum_i (\langle \hat{a}_i, x_0 \rangle - \hat{b}_i)^2 \leq \sum_i (\langle \hat{a}_i, x \rangle - \hat{b}_i)^2$$

$$\leq \sum_i (\langle a_i, x \rangle - b_i)^2 + O(\zeta)\|a_i - \hat{a}_i\| = \|Ax - b\|_2^2 + O(\zeta)\Phi_{k,1}(A).$$

Hence, the Bernstein inequality ensures that

$$\Pr\left[\left|\sum_{e \in S} w(e)(\langle a_e, x \rangle - b_s)^2 - \|Ax - b\|_2^2\right| \geq \varepsilon(\|Ax - b\|_2^2 + \zeta\Phi_{k,1}(A)))\right]$$

$$\Pr\left[\left|\sum_{e \in S} w(e)(\langle a_e, x \rangle - b_e)^2 - \|Ax - b\|_2^2\right| \geq \varepsilon/2 \cdot (\|Ax - b\|_2^2 + \zeta\Phi_{k,1}(A) + \sum_i v(a_i, x_0))\right]$$

$$\leq \exp\left(-\varepsilon^2 s/8\right).$$

Therefore, using that $s = 8\varepsilon^{-2}\log(1/\delta)$ concludes the lemma. $\square$

## C MORE EXPERIMENTAL DETAILS

**Datasets and models.** We largely follow the dataset and model setup used in DISTIL. The datasets we consider are MNIST, Fashion MNIST and CIFAR10. For the first two we use a neural network with one 128-dimensional hidden layer, and for the last one we use convolutional neural network with three convolutional layers and three dense layers. We train each model for 10 epochs, a batch size of 32, and use Adam optimizer with a learning rate of $10^{-3}$.

**Runtime comparison.** In Figure 3, we show a comparison between the runtimes of our loss- and gradient-based algorithms, and the coreset algorithm of Sener & Savarese (2018). All algorithms were implemented in python using the tensorflow framework and the runtime calculation experiments ran on CPU, on a cloud VM with 24 CPUs and 100GB of RAM. It should be noted that a significant advantage of our loss- and gradient-based sampling is that they can rely on a pre-computed metric and clustering that is not updated during the sampling process. In most applications, this will be a fixed metric generated by an upstream model, that is easy to generate and compute distances. As a result, most of the runtime will be spent running model inferences at the cluster center points.

**Runtime comparison on regression.** In Table 1 we compare the runtimes of computing the leverage scores of a matrix and computing a $k$-medoid clustering (a variant of $k$-median). As the results show, even on matrices with i.i.d. normal entries, the standard $k$-medoid implementation from python's scikit-learn library is significantly faster than computing the leverage scores. This is expected, since leverage score computation takes time $O(\text{nnz} + n^\omega)$, where nnz is the number of non-zeros in the data matrix and $\omega \geq 2$ is the runtime exponent of matrix multiplication, while $O(1)$-approximate $k$-median takes time $O(\text{nnz} + n)$.

| $n$ | 5,000 | 10,000 | 20,000 | 30,000 |
|---|---|---|---|---|
| Leverage runtime (sec) | 36.62 | 250.64 | 1890.73 | 6098.31 |
| $k$-medoid runtime (sec) | **19.74** | **105.29** | **858.5** | **2712.09** |

Table 1: Comparing the runtime for computing all leverage scores and a $k$-medoids clustering for $k = 0.2n$, of an $n \times n$ matrix with i.i.d. standard normal entries.

**Algorithms for data selection.** We list some of the best-performing algorithms in literature, and mention their advantages and disadvantages. Out of these, margin and entropy sampling are the top performing methods in the DISTIL[2] benchmark.

---

[2]https://github.com/decile-team/distil

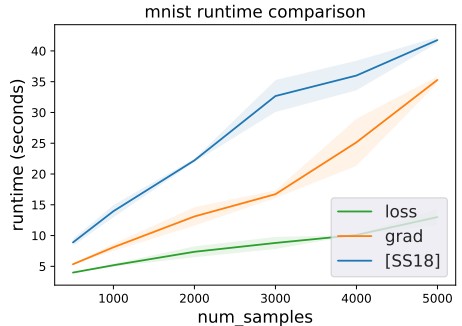 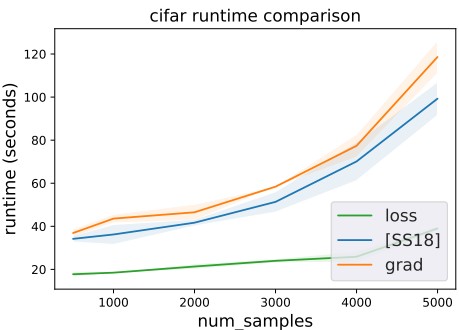

Figure 3: We present runtime comparisons between different algorithms, for MNIST and CIFAR10. The results for Fashion MNIST are analogous to those of MNIST, since the model and dataset have the same size.

- Uniform sampling: We uniformly sample data points up to the budget $k$. This is the simplest and fastest way to sample $k$ data points.
- Margin/Least confidence/Entropy sampling: These methods aim to select the examples with the lowest confidence. Specifically, if $p_1, \ldots, p_C$ are the per-class output probabilities of the model, we select the data points that either minimize $\max_{i \in [C]} p_i$, minimize $p_{i^*} - \max_{i \in [C] \setminus \{i^*\}} p_i$, where $i^* = \text{argmax}_{i \in [C]} p_i$, or maximize the entropy $-\sum_{i=1}^{C} p_i \log p_i$. Unfortunately, these methods require an inference call for *each* data point, in order to evaluate its the classification uncertainty, and so are not considered runtime efficient.
- $k$-center CoreSet [SS18]: The $k$-center algorithm from Sener & Savarese (2018). This algorithm does not require any model inferences, but instead requires maintaining a nearest-neighbor data structure under insertions of data points.

In Figure 4 we provide more detailed experiments, including multiple algorithms from previous work.

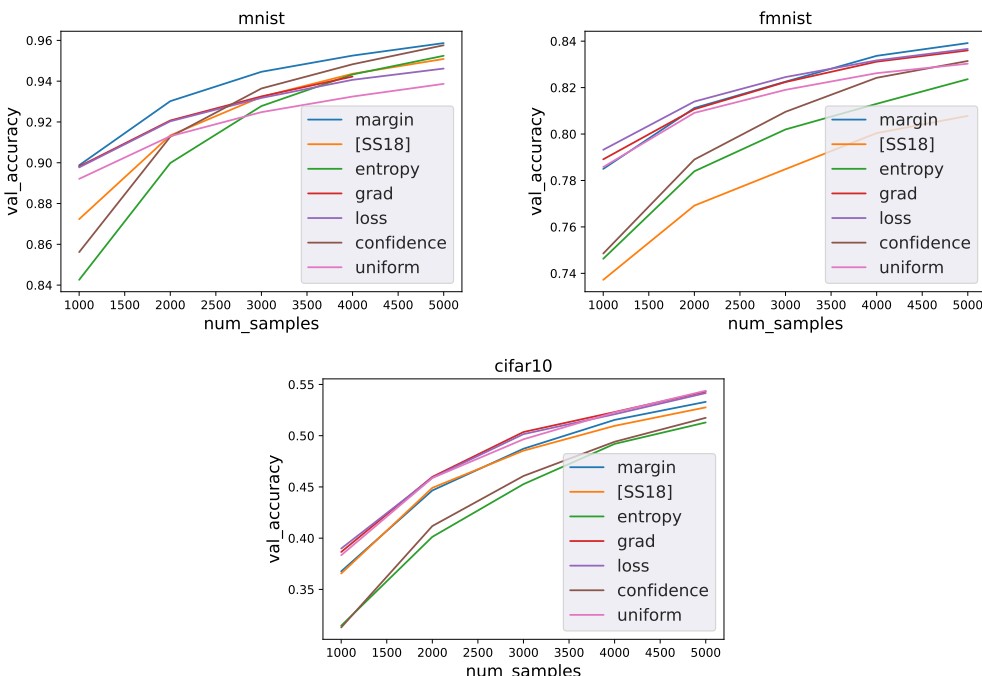

Figure 4: The experimental results for different datasets and algorithms. To minimize variation, we independently run each data point 100 times.

