# OpenReview forum: "Sensitivity Sampling for Coreset-Based Data Selection"
_ICLR.cc/2024/Conference — Submitted to ICLR 2024_

### Official Review · Reviewer_eXnh · 2023-10-30

**Soundness:** 3 good
**Presentation:** 2 fair
**Contribution:** 2 fair
**Rating:** 6
**Confidence:** 2

**Summary:**

This paper proposes a new data selection approach under the Holder continuity assumption. The proposed approach can find a 1/a^2-size coreset with k query to the model, with (1+a) multiplicative error and an additive term. Compared with other approaches, the approach is query efficient because it only needs k inferences from the model. Several experiments are performed to verify the validity of the approach.

**Strengths:**

This paper propose a simple yet query efficient approach. The proof is clear and easy to follow. The proposed approach performs better on several datasets, compared with some previous works.

**Weaknesses:**

1. The Holder continuity assumption lacks empirical support. Does the dataset really satisfy the assumption? It would be better to perform experiments to verify it.
2. The paper lacks comparison with previous works, both theoretically and empirically.

   Theoretically, the proposed method can find a 1/a^2-size coreset with k query to the model, with (1+a) multiplicative error and an additive term, under the Holder continuity assumption. What are the assumptions in previous works? How large is the coreset in previous works? How many are the queries of previous works? I can't find explicit comparisons. It would be better to add a table to list them.

   Empirically, there is only one compared method [1], which is published in 2018. It seems there are several methods for coreset after 2018, as listed in the related work. But there is no comparison with them.
3. It seems there are some errors in the proof and algorithm.

   a. By using Holder continuity of $l$, you obtain that $l(e)\leq \hat{l}(e)+\lambda v(e)$, where $v(e)$ is defined as $||e-A(e)||^z$. It seems $e$ and $A(e)$ belong to the sample space. But the Holder continuity needs that $e$ and $A(e)$ belong to the embedding space. To satisfy the Holder continuity, $v(e)$ should be $||v_e-v_{A(e)}||^z$, where $v_x$ stands for the embedding of $x$. There should be some modifications in proof and algorithm.

   b. It seems that the expectation of $X_i$ (defined in Appendix B.2) is $\sum l(e)w(e)p(e)=\sum l(e)/s$. Therefore, by using Bernstein’s inequality, $|\sum l(e)/s-\sum w(e)l(e)|$ is bounded, rather than $|\sum l(e)-\sum w(e)l(e)|$.
4. There is an extra time cost to obtain the embedding for each sample, which may take about the same amount of time as inference.  Although query efficient, the total time cost may be similar with previous works.


[1] Ozan Sener and Silvio Savarese. Active learning for convolutional neural networks: A core-set approach.

**Questions:**

1. Is Holder continuity a general assumption? Does Holder continuity hold in real datasets?
2. Is [1] the state-of-the-art method? Why not compare with other methods which listed in the related works?
3. How much is the time cost to obtain the embedding of the sample? Does previous work need to obtain the embeddings?

[1] Ozan Sener and Silvio Savarese. Active learning for convolutional neural networks: A core-set approach.

---

> ### Author Response · Authors · 2023-11-16
> **Response to the first question**
>
> We ran experiments on the data and the Lipschitz constant for MNIST is 0.02. We are computing it for the other datasets and will add this to the draft.
> We note that assuming Lipschitzness of the loss function is a common assumption, done e.g. by Sener and Savarese. Since Hölder-continuous for z=1 is equivalent to Lipschitz continuous, our assumption is weaker than that.

---

> ### Author Response · Authors · 2023-11-16
> **Response to the second question**
>
> We would like to stress that we focus on the regime where we can only make, say, k queries to the model to obtain the loss / gradient / margin scores of a subset of k elements, before sampling the rest. It seems that there is very little work in this mixed regime which provides a limited amount of information from the model. Most prior work either uses very indirect model information (updated embeddings like [1], no use of margin or loss scores), or model information for all the input elements (such as margin scores). Our idea was to provide comparison with the two other extreme methods (indirect information vs information for all elements).
> It seems that none of the other methods would provide comparison in the mixed model.

---

> ### Author Response · Authors · 2023-11-16
> **Response to the third question**
>
> Previous work such as Sener and Savarese also require embeddings.
>
> Note that in many cases, pre-computed embeddings are already provided for the input data.
>
> We believe that since it works for these (as shown in our experiments), then this approach can also work more broadly. Indeed, in many cases, one can obtain generic embeddings that would be good enough. One may for example think of BERT embeddings for text applications. Of course the model embeddings could be more refined but elements that are similar under generic embeddings such as BERT should hopefully be similar in the model embeddings too.

---

> ### Author Response · Authors · 2023-11-16
> **Response to the correctness issue raised**
>
> Weakness 3.a: Indeed, the assumption we use is that the loss is Holder-continuous w.r.t the embeddings of the points, thank you very much for catching this. If the loss is Holder-continuous in the sample space, then we do not even need to use the embedding and the pre-trained model, and directly cluster in the sample space. However, this assumption looks much stronger than the Holder continuity in the embedding space.

---

> ### Author Response · Authors · 2023-11-20
> **Response to the correctness issue raised (Weakness 3.b)**
>
> The expectation of $X_i$ is indeed $\sum \ell(e) / s$, but we apply Bernstein’s inequality to $\sum_{i=1}^s X_i$, whose expected value is therefore $\sum \ell(e)$ as desired. We apologize for the confusion

---

> ### Author Response · Authors · 2023-11-22
> **Comparison with previous work (weakness 4 and question 3)**
>
> We would like to mention that previous work that tackle large input data, such as for example the work of Citovsky, DeSalvo, Gentile, Karydas, Rajagopalan, Rostamizadeh,  Kumar (Batch Active Learning at Scale) required to compute the margin scores of all the data element. Similarly previous work on active regression all required leverage score computations which are slower and less accurate on some datasets.

---

> > ### Comment · Reviewer_eXnh · 2023-11-22
> > **Thanks for the response**
> >
> > I have read the answers to the questions and improved my score. However, I think the paper needs more novelty for strongly acceptance.

---

> > > ### Author Response · Authors · 2023-11-22
> > > **Thanks**
> > >
> > > We thanks the reviewer for their time and valuable comments.

---

### Official Review · Reviewer_mfTM · 2023-10-31

**Soundness:** 3 good
**Presentation:** 2 fair
**Contribution:** 3 good
**Rating:** 6
**Confidence:** 3

**Summary:**

This paper provides a coreset-based data selection algorithm that relies on clustering, specifically the $(k,z)$-clustering.

**Strengths:**

In what follows, I will present the strengths of the techniques proposed in the paper:
  * The idea of using $(k,z)$-clustering is interesting, as it is a better candidate than $k$-center for small $z$, in terms of robustness against outliers.
  * The proofs given in the paper are elegantly written, and easy to follow.
  * Connecting the use of H$\ddot{o}$lder Continuity to coresets specifically through the $(k,z)$-clustering problem is innovative.
  * The proposed coreset is small in size which is an advantage, however, with such a small size, an additive approximation term is guaranteed by the coreset in addition to the usual multiplicative factor approximation that most coresets admit.

**Weaknesses:**

While the paper has quite an impressive set of strengths, the paper suffers from the following weaknesses:
   * The writing could use some polishing.
   * The experimental section lacks more information, to better understand what each experiment aims to show -- specifically in the realm of neural networks.
   * Assumption 10 in the field of linear regression seems too restricting.
   * See my questions below.

**Questions:**

Please address each of the following questions:

* Page 3 -- It is stated that "Therefore, our upper-bound is less robust to outliers"
Should your choice of $(k,z)$-clustering make your bounds more robust against outliers as $z$ get smaller? since as $z$ increases, the behavior of the $(k,z)$-clustering tends to behave more like $k$-center problem.

* Page 4 -- Change "there exists a real-valued constant" to "there exist real-valued constants"

* Page 9 -- It is stated "We round them to the closest data point from the dataset in $\ell_2$ norm": This implies that the centers in practice at least are replaced by the data points from the original data such that their embedding are the closest to the centers in the embedding space. right?

* How do you ensure that your version of H$\ddot{o}$lder Continuity holds in the neural network regime?

* Assumption 10 seems too restricting. Can you elaborate on this?

* Can you put time graphs concerning Figure 1 to further highlight the speed gain one would enjoy when using your approach as opposed to using the leverage score sampling technique?

* In the neural network experiments:
  * How many epochs were used to train the uniformly sampled points of size $k^\prime$?
  * Can you put the runtime for uniform sampling coreset in Figure 2 (a) for better comparison?
  * In Figure 4, it was stated "independently run each data point 100 times", did you mean you ran the experiment 100 times with different sampled sets? If not, then please elaborate.
   * Also concerning Figure 4, can you put the running times to highlight the advantages of your approach better -- while margin is the best, it does require the most time (due to the inference payload it has to pay).

---

> ### Author Response · Authors · 2023-11-16
> **Response to the first question**
>
> Yes, that’s right, if we use the $k$-median objective (i.e.: $(k,z)$-clustering with $z=1$), the approach will indeed be more robust against outliers. Thanks a lot for the remark, we will incorporate it into the draft. In particular, the $k$-median objective is similar to the median, which is robust to corruption of 50\% of the dataset ; while $k$-means is much more sensitive to outliers, and $k$-center even worse. Note that the result of Sener and Savarese uses the k-center objective and this may be a reason why we are also able to improve upon their work.

---

> > ### Author Response · Authors · 2023-11-16
> > **Response to the second question**
> >
> > Done, thanks a lot, we will upload a revised version as soon as possible.

---

> ### Author Response · Authors · 2023-11-16
> **Response to the third question**
>
> That’s correct, we will clarify this. Replacing the centers by their closest input point induces a factor $2^z$ loss in the $(k,z)$-clustering cost.

---

> ### Author Response · Authors · 2023-11-16
> **Response to the fourth question**
>
> We have no way of enforcing it. We postulate that it holds. We are conducting experiments that we will add to the current paper as soon as the results are available.
> We note that assuming Lipschitzness of the loss function is a common assumption, done e.g. by Sener and Savarese. Since Hölder-continuous for z=1 is equivalent to Lipschitz continuous, our assumption is weaker than that. We ran experiments on the data and the Lipschitz constant for MNIST is 0.02, we are running it on the other datasets and will update the draft.

---

> ### Author Response · Authors · 2023-11-16
> **Response to the sixth question**
>
> Regarding the runtimes, we will try to run some runtime experiments during the rebuttal period to collect runtimes and update the manuscript as soon as possible.

---

> ### Author Response · Authors · 2023-11-16
> **Response to the seventh question**
>
> The initial model using the uniformly sampled points was trained for 10 epochs. Similarly we will update the manuscript as soon as possible with the running time comparisons.
> As for the question regarding independent runs in Figure 4, that’s exactly right: We run each sampling algorithm 100 times independently, each time sampling different points. We will make this clearer in the new version.

---

> ### Author Response · Authors · 2023-11-17
> **Response to the fifth question**
>
> We have run some experiments to check whether the assumption holds in practice. For the gas sensor dataset:
>
> $||a_i||_2 \leq 300$ for all i's and $\leq 15$ for 90% of i's
>
> $|b_i| \leq 5$ for all i's and $\leq 0.9$ for 90% of i's
>
> $|b_i - b_{​N(i)}| / ||a_i - a_{​N(i)}||_2 \leq 0.87$ for all i and $\le 0.2$ for 90% of i's.
>
> Thus the Lipschitzness condition (Assumption 10) holds.

---

> ### Author Response · Authors · 2023-11-22
> **Runtime experiments for regression**
>
> We have done those runtime experiments on artificial matrices (with i.i.d normal entries), see section C of the appendix. On those examples, the speed-up using our algorithm is roughly a factor 2 compared to computing leverage scores.
>
> We would be happy to answer any further question.

---

### Official Review · Reviewer_MWNf · 2023-10-31

**Soundness:** 2 fair
**Presentation:** 1 poor
**Contribution:** 2 fair
**Rating:** 6
**Confidence:** 3

**Summary:**

The authors propose an application of sensitivity sampling to obtain coresets for clusterable data. Their method involves computing approximate k-means or k-median clusters over a dataset, and selecting data points based on a modified sensitivity sampling method, where each point is chosen with probability proportional to its distance from the centers and the loss associated with the center. Their results differ from previous methods because they study a class of losses more general than Lipschitz functions, but a subset of Holder continuous functions.

Their main theoretical contribution lies in showing that with constant probability, the weighted loss on their proposed coreset closely approximates the loss on the entire dataset, with an additive error dependent on the cost of clustering the data.

They provide experiments where they compare their method to existing sampling techniques like uniform sampling, leverage score sampling, other clustering based methods for tasks like linear regression and training neural networks on datasets, including gas sensors and CIFAR.

**Strengths:**

Their contribution of applying the preprocessing step to obtain $(k,z)$ clustering is interesting since the samples are diverse and not sensitive to outliers. Moreover, their results extend to tasks beyond classification because of the loss being Holder continuous. Empirically, their method seems to perform similar to existing methods for linear regression and neural network based tasks in the low sample regime.

**Weaknesses:**

Addressed in the questions section.

**Questions:**

* For the case of linear regression, it is not clear how the proposed method is theoretically better (in terms of number of samples or runtime) than existing works using sensitivity sampling including Chen and Price, Chen and Derezinski, etc. Moreover there are additional assumptions on the data as opposed to distributional assumptions which is not stated clearly. It would be good to have a detailed explanation for the assumption 10 that requires for any $i,j$ $| b_i - b_j | \le ||a_i - a_j||_2$.

* It seems that their primary theoretical contribution revolves around the use of Holder-continuous loss functions; however, their experiments do not seem to leverage this property.

* This paper as a whole is not reader friendly and requires significant time commitment because of imprecise definitions, lack of explanations and inconsistent terminology. For example, Algorithms 1 and 2 are never referred. The definition of $\mathcal{A}$ from the first lines of both algorithms is not properly stated and it is also overloaded $\mathcal{A}(e)$. Is it defined as $\mathcal{A}(\mathcal{D}) := \min_{|C| = k} C' \cdot \Phi_z(\mathcal{D},C)$ for some constant $C'$? In the clustering objective $C$ is not defined correctly where $|C| = k$ and $C \subset \mathbb{R}^d$. There is also inconsistency in using norms, somewhere $||\cdot||$ or $||\cdot||_2$ is used. $\Delta$ is mentioned in section 3.1 without definition. Overall I recommend a better use of space for defining variables as opposed to other sections like stating Berstein's inequality.

 Given these questions, their overall contribution appears reasonable in terms of the empirical study of their proposed algorithm, but the theoretical contribution is lacking.

---

> ### Author Response · Authors · 2023-11-16
> **Response to the first weakness bullet**
>
> The runtime of our method is better than the runtime of computing leverage score in theory for high-dimensional input: computing a $k$-median solution can be done in time essentially nnz $+ n/\epsilon^2$, while the state-of-the-art methods for computing the leverage scores require nnz + $d^\omega$ (where nnz is the number of non zero elements in the description of the input, and $\omega$ is the exponent of matrix multiplication). We will soon provide an experimental evaluation to complete the theoretical analysis.

---

> > ### Author Response · Authors · 2023-11-22
> >
> > We have done those runtime experiments on artificial matrices (with i.i.d normal entries), see section C of the appendix. On those examples, the speed-up using our algorithm is roughly a factor 2 compared to computing leverage scores.
> >
> > We would be happy to answer any further question.

---

> > > ### Comment · Reviewer_MWNf · 2023-11-22
> > >
> > > Thanks for adding the experiments, while I still lean toward acceptance, I agree with the other reviewers that the contribution is lacking in terms of the novelty for a strong acceptance. I retain my score for this paper.

---

> ### Author Response · Authors · 2023-11-16
> **Response to the second weakness bullet**
>
> This is a very fair point. We ran experiments on the data and the Lipschitz constant for MNIST is 0.02 and so indeed bounded. We will also provide plots to evaluate the Lipschitz constants on the remaining datasets we used in the updated version of our paper. To be clear, our algorithm does use the Holder-continuity property in the definition of the sampling probabilities (it is the lambda parameter in Algorithm 1).

---

> ### Author Response · Authors · 2023-11-16
> **Response to the third weakness bullet**
>
> We apologize for the general lack of clarity,  we are working on an improved version that we will upload as soon as possible. To answer your questions, $\mathcal{A}(e)$ is defined in algorithm 1 as $\text{argmin}_{a \in \mathcal{A}} ||a-e||$, the closest center to e in $\mathcal{A}$. We never use the notation $\mathcal{A}(\mathcal{D})$, which would indeed be conflicting with the previous one.
> Algorithm 1 is referred to in the proof of Theorem 6, Algorithm 2 in Theorem 10 but we agree that it is good to refer to it more explicitly.

---

> ### Author Response · Authors · 2023-11-16
> **Response to the final conclusion**
>
> "Given these questions, their overall contribution appears reasonable in terms of the empirical study of their proposed algorithm, but the theoretical contribution is lacking."
>
>
> We would like to point out that our contribution also consists in showing that one can remove several assumptions made by Sener and Savarese in their theoretical framework. The framework we focus on is much crisper, only relying on the Holder-continuity (which we show holds in practice) and so we can obtain sampling algorithms that are both simpler and achieve better results in practice. Thus, we believe that the new theoretical framework we provide is a new framework for developing algorithms with theoretical guarantees.

---

### Official Review · Reviewer_nKWs · 2023-11-01

**Soundness:** 3 good
**Presentation:** 2 fair
**Contribution:** 2 fair
**Rating:** 5
**Confidence:** 4

**Summary:**

The paper presents a data selection algorithm for ML models with a holder continuous loss, based on the idea of first doing a $(k,z)$ clustering of the points and then using the clustering to define sampling probabilities for the points. Sampling and reweighing the sampled points appropriately, gives you a subset which can approximate the loss over the full data with an additive error proportional to the clustering cost. The authors specifically show the theoretical guarantees of the algorithm for the case of linear regression. Finally, they perform experiments to validate their claims comparing their sampling technique with uniform sampling and $k$- center based technique for the case of neural networks and leverage scores for regression problem.

**Strengths:**

1. Data selection is an important problem and as such the paper will be of interest to the community.
2. The paper, for the most part, is well written and is easy to follow.
3. Experiments for the Neural network seem good.

**Weaknesses:**

1. I do not feel the paper has enough novelty at all for a venue like ICLR. The idea seems a combination of the one by Sener (2018) and results from coreset literature with only minor or incremental modifications.
2. The paper title has sensitivity; however, the paper does not really use sensitivity as defined in literature. It appears to use a very crude approximation to sensitivity which is why there is an additive error. This idea also is very similar to the one of "light weight coresets".
3. The additive error can be pretty large I believe.
4. The proofs are pretty straight forward (not a bad thing in itself); however, it means even proof technique wise there is not much contribution. For e.g., the proof of theorem 6 is direct application of Bernstein and is well known. The proof for the 1-round algorithm is also very similar to existing proofs.
4. For regression, theoretical guarantees are much weaker than ones obtained using leverage scores. Empirically the authors claim they can match leverage score sampling in much less time. However, I could not find if they report the time for regression experiments.
5. There are small writing errors as well. For e.g.: what is $X$ in theorem 1,7? In Algorithm 2 $x_0$ is mentioned in step 2 and computed in step 3.
Overall, I think the authors need to clarify and highlight the contributions of the paper more and contrast them with existing results, especially in terms of novelty.

**Questions:**

Please try and address the points given as weaknesses.

---

> ### Author Response · Authors · 2023-11-16
> **Response to Weakness #1**
>
> There is a drastic difference with the result of Sener and Savarese which lies in the fact that the approach considered there does not use any direct model information (such as loss, margin score, or gradient). The empirical success of methods such as margin or confidence sampling has shown that model-based data selection can achieve much higher quality results than model-agnostic selection. Our approach allows using model-based scores (e.g. loss) to improve selection quality, while maintaining the efficiency of coreset selection. As such, even in the era of foundation models and big datasets, our approach allows using a few model queries to obtain proxy estimates for margin, loss, or gradient that gives a better sampling procedure.
>
> Our experimental results agree with this intuition, since our approach yields samples of significantly improved quality compared to Sener and Savarese [SS18]. Moreover, Sener and Savarese also makes such an assumption Since Hölder-continuous for z=1 is equivalent to Lipschitz continuous, our assumption is weaker than that.

---

> > ### Author Response · Authors · 2023-11-16
> > **Response to Weakness #2**
> >
> > Sampling according to the $(k,z)$-clustering cost provides an approximation of the sensitivity, as defined in the coreset literature as $\sup_{solution S} \frac{\cost(p, S)}{\cost(P, S)}$. In this literature, this sampling is (perhaps abusively) called sensitivity sampling for that reason. Thanks for pointing out the “Light weight coresets” paper, we will definitely add a citation.

---

> ### Author Response · Authors · 2023-11-16
> **Response to Weakness #3**
>
> The idea is that this error nicely scales with the number of centers chosen, as k grows, the error vanishes (when k=n it is 0). Moreover, the magnitude of this error relative to the actual loss depends on the Lipshitz constant, whenever this constant is of bounded magnitude, the error is bounded compared to the loss.
>
> We ran further experiments and the Lipshitz constant for the MNIST dataset is 0.02. This implies that for this dataset, the additive term is not large compared to the total loss.
>
> More importantly, we demonstrate in practice that this approximation provides competitive results, coming close to very efficient methods (such as margin score) at a much cheaper computational cost.

---

> ### Author Response · Authors · 2023-11-16
> **Response to Weakness #4**
>
> As mentioned in the answer to remark 1, our contribution is mainly in the idea of using the $(k,z)$-clustering scores as a sampling distribution, that serves as proxy distribution to the loss distribution.

---

> ### Author Response · Authors · 2023-11-16
> **Response to Weakness #5**
>
> We would like to note that the use of our approach is not only faster empirically but also theoretically for high-dimensional inputs: computing a $k$-median solution can be done in time essentially nnz $+ n/\epsilon^2$, while the state-of-the-art methods for computing the leverage scores require nnz + $d^\omega$ (where nnz is the number of non zero elements in the description of the input, and $\omega$ is the exponent of matrix multiplication). Regarding empirical results, we will upload a runtime comparison as soon as possible.

---

> ### Author Response · Authors · 2023-11-16
> **Response to Weakness #6**
>
> We apologize for those errors, and will soon post an updated version.

---

> ### Comment · Reviewer_nKWs · 2023-11-21
> **Thanks for the response**
>
> I read the other reviews and your answers to the questions. I have improved my score slightly. However, I still feel the paper needs more clarity and novelty for acceptance.

---

> > ### Author Response · Authors · 2023-11-22
> > **Thanks!**
> >
> > We thank the reviewer for their time and careful reading. We have uploaded a new version of our manuscript where we have improved the writing quality and incorporated the comments made by all the reviewers, hoping that this will make it clearer.

---

### Meta-Review · Area_Chair_ypKv · 2023-12-15

**Metareview:**

This paper describes a new active learning approach for efficiently selecting training data in the context of growing data and model sizes. This method uses k-means clustering and sensitivity sampling, based on data embeddings and assuming a Lipshitz-continuous model loss. It is proven to select a subset of data whose average loss closely approximates the entire dataset's loss, with high efficiency requiring minimal model inferences. The approach's effectiveness is demonstrated on classic datasets, where it outperforms existing state-of-the-art methods.

This is a borderline paper. Most reviewers were borderline and pointed out that while this paper does make some contributions from a theoretical and algorithmic perspective, the results were incremental, and the bounds would be likely not as strong. I agree with these issues. I would recommend the paper address the issues pointed out the reviewers.

**Justification For Why Not Higher Score:**

This paper would benefit from a revision before it is at a level to be resubmitted.

**Justification For Why Not Lower Score:**

N/A

---

### Decision · Program_Chairs · 2024-01-16

Reject